# Combined laser and ozone therapy for onychomycosis in an *in vitro* and *ex vivo* model

**Javier Fernández**[1,2,3], **Iván del Valle Fernández**[4], **Claudio J. Villar**[1,2,3], **Felipe Lombó**[1,2,3]*

**1** Departamento de Biología Funcional, Research Unit "Biotechnology in Nutraceuticals and Bioactive Compounds-BIONUC", Área de Microbiología, Universidad de Oviedo, Oviedo, Spain, **2** Instituto Universitario de Oncología del Principado de Asturias, Oviedo, Spain, **3** Instituto de Investigación Sanitaria del Principado de Asturias, Oviedo, Spain, **4** Termosalud SL, Polígono Industrial Los Campones Tremañes, Gijón, Asturias, Spain

* lombofelipe@uniovi.es

**Data Availability Statement:** All relevant data are within the paper and its Supporting information files.

**Funding:** This study was supported by the Programa Ayudas a Empresas para la Ejecución de

## Abstract

In order to develop a fast combined method for onychomycosis treatment using an *in vitro* and an *ex vivo* models, a combination of two dual-diode lasers at 405 nm and 639 nm wavelengths, in a continuous manner, together with different ozone concentrations (until 80 ppm), was used for performing the experiments on fungal strains growing on PDA agar medium or on pig's hooves samples. In the *in vitro* model experiments, with 30 min combined treatment, all species are inhibited at 40 ppm ozone concentration, except *S. brevicaulis*, which didn't show an inhibition in comparison with only ozone treatment. In the *ex vivo* model experiments, with the same duration and ozone concentration, *A. chrysogenum* and *E. floccosum* showed total inhibition; *T. mentagrophytes* and *T. rubrum* showed a 75% growth inhibition; *M. canis* showed a delay in sporulation; and *S. brevicaulis* and *A. terreus* did not show growth inhibition. This combined laser and ozone treatment may be developed as a fast therapy for human onychomycosis, as a potential alternative to the use of antifungal drugs with potential side effects and long duration treatments.

## Introduction

Onychomycosis is a nail infection caused by fungus or yeast [1]. The clinical symptoms of this disease are discoloration of the nails, painless detachment of the nail bed (onycolysis), and hyperkeratosis. These symptoms can reduce the patient's quality of life with pain, paresthesia, stress or making social relationships difficult [2,3].

Onychomycosis is the most common nail disease, with a prevalence of 5.5% worldwide. Diverse risk factors increase the probability of suffering this disease, such as age, trauma, disease (diabetes, obesity, etc.), immunosuppression, psoriasis (56% increased risk), *tinea pedis* infection, genetics (associated to genes such as *mhc* or *hla-dr8*), direct transmission from an infected person or lifestyle habits (sports, smoking or the type of shoes) [4–7].

The main fungi causing onychomycosis are dermathophytes (60%–70%), especially 3 species: *Trichophyton rubru*m (>50%), *T. mentagrophytes* (20%) and *Epidermophyton floccosum*.

Proyectos de I+D+i en el Principado de Asturias en el Periodo 2014-2015 (IE-14-084) and the Programa de Ayudas a Grupos de Investigación del Principado de Asturias (IDI/2018/000120) in the form of grants to FL. IVF received salary from Termosalud SL. The specific roles of these authors are articulated in the 'author contributions' section. The funders had no role in the study design, data collection and analysis, decision to publish, or preparation of the manuscript.

**Competing interests:** The authors have read the journal's policy and have the following competing interests: IVF is paid employee of Termosalud SL. There are no patents, products in development or marketed products associated with this research to declare. This does not alter our adherence to PLOS ONE policies on sharing data and materials. All other authors declare that they have no competing interests.

Dermathophytes are responsible for 90% of toenail and 75% of fingernail onychomycosis. Non-dermathophyte molds cause 20%-30% of nail infections, mainly in toenails, and usually after previous nail traumas. These pathogens belong to genera such as *Fusarium*, *Aspergillus*, *Acremonium*, *Scytalidium* and *Scopulariopsis* [3,8,9]. Finally, yeasts cause 10% to 20% of all onychomycosis, with main species being *Candida albicans* (accounting for 70% of all yeast onychomycosis), *C. tropicalis* and *C. parapsilosis*, mainly in immunosuppressed patients or those ones with vascular problems [10–13].

Diagnosis is important for selecting the right treatment. A common diagnosis uses potassium hydroxide to dissolve nail keratin and to observe the sample under the microscope. This test allows us to discriminate between dermatophytes and saprophytes, although it is only 60% sensitive. Another more sensitive diagnostic test is the nail biopsy embedded in acid-Schiff plus Grocott's methenamine silver stain. It is often necessary to culture the fungus in order to know the specific causal agent, although this process is also not very sensitive (60%) [14,15]. Another method is the PCR technique, which can determine the causal agent by amplification of 18S rRNA. All of these techniques are not 100% effective, so a combined use allows a better identification, decreasing false negatives [16–18].

Main treatments for onychomycosis are oral or topical. Oral treatments (fluconazole, terbinafine, etc.) are used in moderate or severe onychomycosis, but these treatments may cause liver side effects, drugs interference and other health problems such as heart failure, proteinuria, photosensitivity, and intestinal disorders. Topical treatments (such as K101 solution, efinaconazole, etc.) are used in milder cases or in those cases where oral treatments are not recommended. These topical treatments often show low penetration through the nail. Also, they may cause local disorders, such as dermatitis. Both types of treatments are usually long term (over 3 to 9 months) and ineffective in some cases of onychomycosis [19–22].

There are new treatments using physical methods, such as lasers, photodynamic therapy or iontophoresis that are beginning to be used in these last years, although many of them are still in research phase [23–25]. One of the advantages of laser therapies is their ability to concentrate the energy beam on the affected tissue, thus reducing possible side effects, such as those that exist with oral treatments. All lasers currently approved by the FDA for the treatment of onychomycosis are based on neodymium-itrium-aluminum (Nd:YAG 1064 nm), $CO_2$, and femtosecond infrared titanium-sapphire [26,27]. These photodynamic therapies are based on the specific absorption of the laser energy by fungal chromophores which are not present in the human tissues, therefore causing a photothermic and photochemical effect, destroying the fungal hyphae, depending in main parameters such as wavelength, pulse duration, frequency, irradiation area, and the number and timing of the treatments. This absorption thus causes photothermal and photochemical effects that destroy the hyphae of the fungus, without causing systemic effects [28,29]. Specifically, the violet 405 nm wavelength has been described as causing oxidative stress in fungal cells, targeting nicotinamide adenine dinucleotide phosphate-oxidase (NADPH-oxidase) and cytochrome C oxidase, increasing its production of the highly volatile reactive oxygen species (ROS), which have antimicrobial effects. On the other hand, the red wavelength (639 nm) is used to activate immune cells and increase microcirculation at the nail bed tissue, reinforcing the recruitment of immune cells (neutrophils) to this infected area. Parameters such as laser wavelength, duration of the light pulse, frequency, area of irradiation, and the number and time spacing between treatments affect the effectiveness of these therapies. Another alternative is the combination of laser treatment together with photosensitizers, such as porphyrins, in order to enhance the generation of free radicals in the nail substrate [30–33].

Ozone ($O_3$) is a potent oxidizing gas, widely used as antimicrobial agent. This gas has been also recently used as a promising onychomycosis treatment option, used at several medical

skin treatments at concentrations over 100 parts per million (ppm) [34–36]. Its antimicrobial activity is widely demonstrated, based on its pro-oxidant effect on microbial cells, sometimes as ozonated oil treatments [37].

The widespread use of laser therapy still needs *in vitro* and *ex vivo* tissue studies such as those proposed in this research project, where its treatment alone or in combination with ozone will be compared with the effectiveness of conventional drugs.

In our study, it was performed an *in vitro* and *ex vivo* assay against the eight most frequent species of fungi in human onychomycosis. This study has been carried out with a prototype equipment that combines the use of low-level laser therapy (LLLT) and ozone. The purpose of this equipment is to carry out a universal fast treatment for the main fungi species that cause onychomycosis.

## Materials and methods

### Equipment

The equipment is composed of a 8.9 cm x 12.4 cm chamber (110.36 cm$^2$) which receives dual-diode laser power (1.8 W/110.36 cm$^2$, which equals 16.3 mW/cm$^2$) at 405 nm (min. 400 nm, typ. 405 nm, max. 410 nm) and 639 nm (min. 635 nm, typ. 639 nm, max. 645 nm) wavelengths in a continuous manner. No thermal effects are generated in this chamber (temperature always below 30 ˚C in the presence of laser).

The equipment also contains an ozone generator able to produce up to 80 ppm ozone. This gas is pumped via a tube into a closed transparent bag (transmittance of 94.9% at 405 nm laser and 96.5% at 639 nm laser) placed around the foot or sample to be treated. This bag is placed in the chamber, below the laser diodes (Fig 1).

### Fungal strains collection

Eight pathogenic fungal species involved in human onychomycosis were obtained from the American Type Culture Collection (ATCC), and were maintained on Potato Dextrose Agar (PDA) culture medium (potato starch: 4 g/L; dextrose: 20 g/L; agar: 15 g/L). The fungal species are: *Acremonium chrysogenum* ATCC 14615, *Aspergillus terreus* ATCC 1012, *Epidermophyton floccosum* ATCC 26072, *Fusarium solani* ATCC 46939, *Microsporum canis* ATCC 10214, *Scopulariopsis brevicaulis* ATCC 7123, *Trichophyton mentagrophytes* ATCC 28185, *Trichophyton rubrum* ATCC 22402.

Dense spores' suspensions were obtained from each fungal species using filtration through steril cotton, and they were maintained at 4˚C in distilled water. These suspensions were quantified on PDA medium in order to use 10$^5$ CFUs for the *in vitro* and *ex vivo* experiments. Incubation temperature was 24˚C for all species, except for *A. chrysogenum* (26˚C).

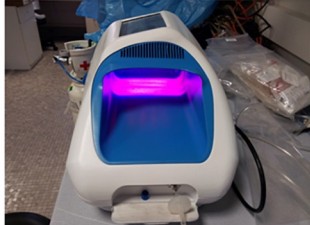 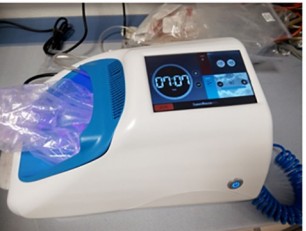

**Fig 1. Equipment designed for the combined ozone and laser therapy of onychomycosis, showing the central irradiated chamber (left) where the bag receiving the ozone gas is placed (right).**

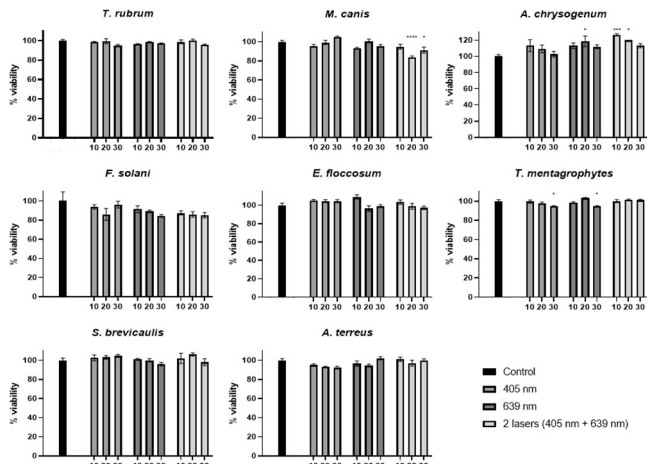

**Fig 2. Percentage of viability for each of the eight fungal species with laser treatments using only 405 nm, only 639 nm and both lasers combined at three different times (10 min, 20 min and 30 min).** Statistical significant results are shown in comparison with the absolute control and indicated with asterisk.

## *In vitro* treatments

For the different ozone concentrations to be tested directly on the fungi cultivated on Petri dishes (PDA medium), these plates were individually placed inside the sealed transparent bag (without the plastic lids). Laser treatments were carried out using only 405 nm, only 639 nm and both lasers combined (Fig 2). Also, ozone (alone) concentrations of 20 ppm, 40 ppm, 60 ppm and 80 ppm, at 10 min, 20 min or 30 min were used (Fig 3). All experiments were carried out in triplicates.

Each Petri dish contained a spores' inoculum of $10^5$ CFUs just before the corresponding ozone treatment or for each control, and the effect of each treatment was monitored after 7 days incubation at the corresponding fungus optimal growth temperature.

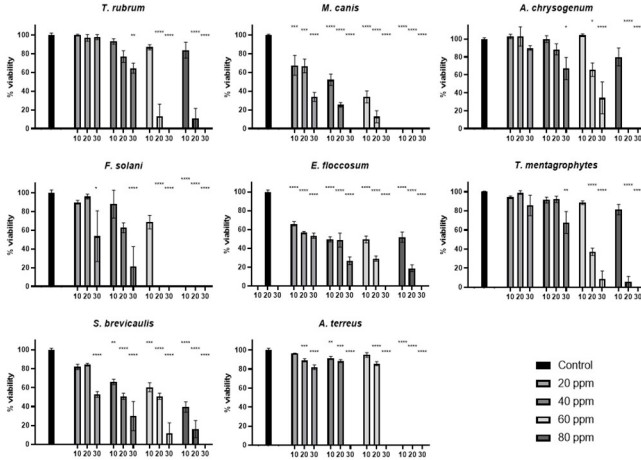

**Fig 3. Percentage of viability for each of the eight fungal species with ozone treatments (20 ppm, 40 ppm, 60 ppm and 80 ppm) at three different times (10 min, 20 min and 30 min).** Statistical significant results are shown in comparison with the absolute control and indicated with asterisk.

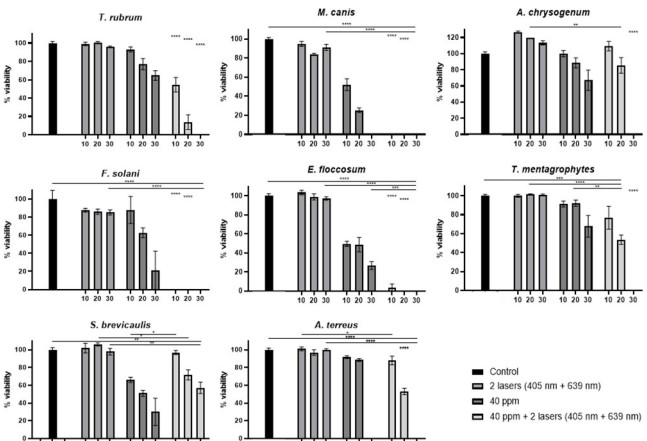

**Fig 4. Percentage of viability for each of the eight fungal species with ozone treatment 40 ppm, in combination with both laser treatments simultaneously at three different times (10 min, 20 min and 30 min).** Statistical significant results of the combined ozone and laser treatments are shown in comparison with only ozone, only laser treatments or absolute control at each time point, and indicated with asterisk. Asterisks without a comparison line indicate the same statistical significant results between all conditions at the same time.

Similar *in vitro* experiments were performed, using 40 ppm or 60 ppm ozone, at 10 min, 20 min or 30 min, but adding during these treatment times a laser irradiation step at 405 nm and 639 nm simultaneously (Figs 4 and 5).

## *Ex vivo* treatments

As a more similar infection model for human onychomycosis, an *ex vivo* model was developed. In this model, pig's hooves samples of about 1 cm$^2$ size and 1 mm thickness were used, which were obtained with permission from the slaughterhouse *Matadero Central de Asturias* (Noreña, Spain) for using them in this research project. This *ex vivo* model, which uses tissue waste that usually will be discharged, is a perfect alternative avoiding the use of animal

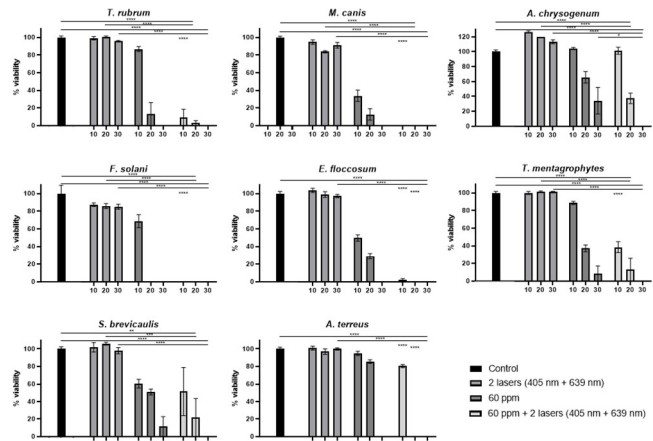

**Fig 5. Percentage of viability for each of the eight fungal species with ozone treatment 60 ppm in combination with both laser treatments simultaneously at three different times (10 min, 20 min and 30 min).** Statistical significant results of the combined ozone and laser treatments are shown in comparison with only ozone, only laser treatments or absolute control at each time point, and indicated with asterisk. Asterisks without a comparison line indicate the same statistical significant results between all conditions at the same time.

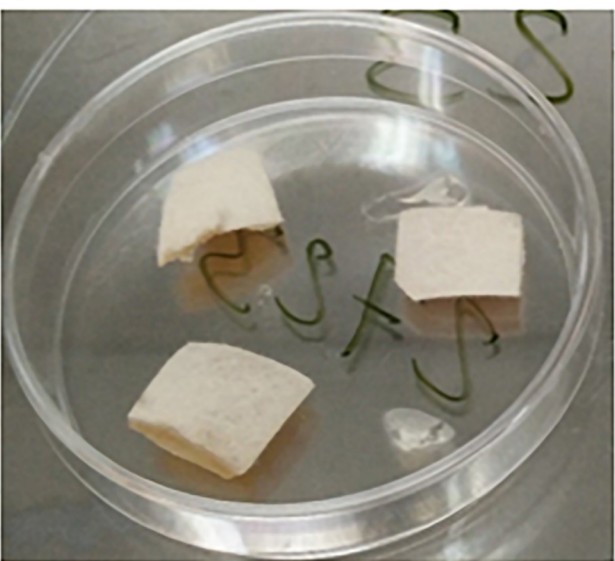

**Fig 6. Porcine hoof samples (1 cm$^2$) inoculated with 10$^5$ CFUs of one of the fungal strains, after 2 weeks incubation period under water-saturated atmosphere.** The white mycelium can be observed on the hoof surface.

experimentation models, and therefore saves an application to the Ethics Committee of the University of Oviedo, as this was not needed. These hooves were autoclaved and then inoculated with 10$^5$ CFUs of the corresponding spore inoculum and were incubated during two weeks in a water-saturated atmosphere (in the middle of a Petri dish) at the corresponding temperature of each fungal strain (Fig 6).

After these two weeks of incubation time, a homogeneous mycelial growth was clearly observed on the surface of each inoculated hoof sample. Then, the combined ozone (40 ppm or 60 ppm) and laser treatment (405 nm and 639 nm simultaneously) was carried out during 30 min. After the corresponding treatment, the hoof sample was embedded in the center of a PDA plate, and incubated for 7 days at the corresponding temperature, in order to test the eventual inhibition of the mycelial growth in the Petri dish after the treatment.

## Statistical analysis

Data were expressed as the mean value ± S.E.M. Statistical analyses were conducted using ANOVA test (Tukey's multiple comparisons test or Dunnett's multiple comparisons test). Normality was analysed using Shapiro-Wilk. The graphical representation of all these data was generated using GraphPad Prism software (version 9.1.0., GraphPad Software, San Diego, CA, USA). In all cases, a p value < 0.05 was considered statistically significant ($^*$p < 0.05; $^{**}$ p < 0.005; $^{***}$ p < 0.0005; $^{****}$ p < 0.0001).

## Results

### Effect of laser treatments on fungal growth *in vitro*

The two different laser wavelengths, 405 nm and 639 nm, were tested individually and simultaneously, on solid medium (triplicates) inoculated with the different fungal species. Three different treatment times were tested (10 min, 20 min and 30 min). No statistically significant differences, regarding the percentage of viability, were observed in the case of *T. rubrum*,

*F. solani*, *E. floccosum*, *S. brevicaulis* nor *A. terreus* (Fig 2 and S1 Rawdata). However, in the case of *M. canis*, a statistically significant viability reduction (in comparison with control experiment) was observed at 20 min (16.2% reduction) and 30 min (8.9% reduction) treatment with both lasers together (Fig 2). In the case of *T. mentagrophytes*, the only significant reduction in viability was observed in the case of 30 min treatment with both lasers separately, and in both cases this viability reduction was 5% (Fig 2). Finally, in the case of *A. chrysogenum*, a 18% increase in fungal growth was observed in the case of 20 min treatment with the 639 nm laser, and also an increase was observed with both lasers applied simultaneously during 10 min (26% increase) and 20 min (20% increase) (Fig 2).

## Effect of ozone treatments on fungal growth *in vitro*

Four different ozone concentrations (together with a control one at 0 ppm) were used for treating inoculated Petri dishes with the corresponding fungal strains: 20 ppm, 40 ppm, 60 ppm and 80 ppm. Treatment durations were 10 min, 20 min or 30 min.

 *T. rubrum*, *A. chrysogenum* and *T. mentagrophytes* show a statistically significant viability reduction only with 40 ppm ozone during at least 30 min (36% reduction in the case of *T. rubrum*, 32.9% reduction in *A. chrysogenum*, 32.1% reduction in *T. mentagrophytes*) (Fig 3).

 In the case of *F. solani* and *S. brevicaulis*, 20 ppm ozone must be applied for at least 30 min (46.5% and 47.2% reduction respectively) in order to cause a reduction in viability. In these two species, 60 ppm during 20 min (*F. solani*) or 80 ppm during 30 min (*S. brevicaulis*) causes total death (Fig 3).

 *A. terreus* shows a statistically significant viability reduction (11.7%) with 20 ppm during 20 min and total inhibition with 40 ppm during 30 min (Fig 3). *M. canis* and *E. floccosum* are more sensitive, as their viability (32.4% and %, 33.9% respectively) is quite reduced already at 20 ppm during only 10 min (Fig 3).

## Effect of combined 40 ppm or 60 ppm ozone and laser treatments on fungal growth *in vitro*

40 ppm ozone concentration was used for treating inoculated Petri dishes with the corresponding fungal strains, together with both laser treatments simultaneously. This treatment was compared with an absolute control (no treatment) and with 40 ppm treatment or both laser wavelengths (405 nm and 639 nm) treatment alone. Treatment durations were 10 min, 20 min or 30 min. All these Petri dishes were treated just after inoculation with the corresponding spores.

 *T. rubrum*, *M. canis*, *F. solani*, and *E. floccosum* show a statistically significant viability reduction when laser treatment is combined with 40 ppm ozone, in comparison with only ozone treatment. This positive effect in viability reduction is already observed in the 10 min treatment (Fig 4). *T. rubrum* shows a 38.5% reduction with 10 min combined treatment, with respect to ozone treatment alone. Over 95% viability reduction is observed with the combined treatment at 10 min, in the case of the last three species (Fig 4). In the case of *T. mentagrophytes* and *A. terreus*, 20 min of combined treatment are necessary in order to observed a statistically significant viability reduction (38.4% and 35.5% respectively) in comparison with ozone treatment alone (Fig 4). Regarding *A. chrysogenum*, 30 min of combined laser and ozone treatment is necessary for observing a significant reduction (total inhibition) in comparison with only ozone treatment (Fig 4). Surprisingly, in *S. brevicaulis*, the inhibitory effect caused by ozone is counteracted when laser is applied (Fig 4).

### Effect of combined 60 ppm ozone and laser treatments on fungal growth *in vitro*

Also, 60 ppm ozone concentration was used for treating inoculated Petri dishes with the corresponding fungal strains, together with both laser treatments simultaneously. As in the previous case, this treatment was compared with the corresponding controls. In all these treatments, In the case of *M. canis*, *F. solani* and *E. floccosum*, 10 min of combined treatment achieves a growth inhibition greater than 90% (Fig 5). In the case of *T. rubrum*, the effect of combined treatment causes a 77.6% viability reduction (almost double inhibition than with 40 ppm and lasers combination). Regarding *A. terreus* and *T. mentagrophytes*, the significant effect of combined treatment is achieved already with 10 min (instead of 20 min required with combined 40 ppm ozone and lasers combination) (Fig 5). *A. chrysogenum* requires 30 min for a significant reduction in viability (combination of 60 ppm and lasers treatment), but this reduction was already observed with the combined treatment using 40 ppm and lasers (Fig 5). Finally, in the case of *S. brevicaulis*, again, the incorporation of lasers in the combined treatments shows no differences with 60 ppm ozone treatment alone (Fig 5).

### Effect of combined ozone and laser treatments on fungal growth *ex vivo*

The effect of a 30 min ozone (40 ppm or 60 ppm) and lasers combined treatment, was different between the fungal strains. In the case of *A. chrysogenum* total inhibiton was already observed at 40 ppm plus lasers. In the case of *E. floccosum*, the total inhibition was obtained at 60 ppm. *T. mentagrophytes* and *T. rubrum* showed a 75% growth inhibition at 60 ppm. In the case of *M. canis*, a delay in the sporulation is observed at 60 ppm, without growth inhibition. *S. brevicaulis* and *A. terreus* do not show growth inhibition in this *ex vivo* model, and *F. solani* was not able to growth on the hooves samples even at 0 ppm ozone control conditions (Fig 7).

## Discussion

Non-thermal laser strategies have been used for treating onychomycosis with variable results [30]. In the case of 405 nm lasers, this wavelength targets NADPH-oxidase, an enzyme which is involved, in generation of reactive oxygen species in eukaryotic cells, exerting antifungal effects. After irradiation, fungal structures, such as lipid membranes, accumulate higher levels of oxidation species, such as malondialdehyde, and at the same time, important antioxidant enzymes, such as superoxide dismutase, are depleted in the treated hyphae. The deleterious effects of laser treatment can be enhanced if photosensitizers, such as porphyrins, are incorporated [31,33].

The other laser used in this study, 639 nm red wavelength, activates immune cells and increases blood circulation in the subungual bed, enhancing the immune response against pathogens such as fungal cells. This enhancement seems to be due to activation of vascular endothelial growth factor, VEGF. These antifungal effects have been already demonstrated against a variety of fungal pathogens, such as the dermatophyte *Trichophyton* spp. [32,38,39].

In this study, these two laser wavelengths (405 nm and 639 nm) have been tested (separately and together) in an *in vitro* onychomycosis model, against eight fungal pathogens. Three treatment times were tested with these lasers, 10 min, 20 min and 30 min, but as a general rule, no effect in the fungal viability was observed with these treatments. Only in two cases (*M. canis* and *T, mentagrophytes*) a very small reduction (5% to 16%) in fungal growth was observed (Fig 2). The reason for this discrete effect on fungal growth may be the short duration of our treatment (only one session, with a maximum of 30 min time frame), instead of a treatment with

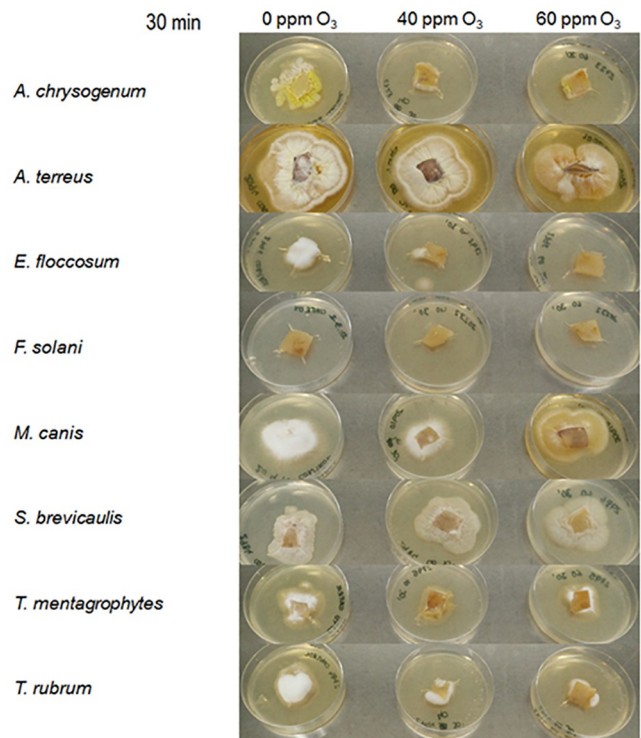

**Fig 7. Growth of the eight fungal strains on porcine hooves samples placed on PDA medium, after ozone treatments at 40 ppm or 60 ppm combined with both laser wavelengths, during 30 min.**

several sessions over several weeks), and also, because the low surface power density used (16 mW/cm$^2$) in comparison with other studies (up to 45 mW/cm$^2$) [32,38,39].

Ozone treatment alone was also tested in this work, against the same eight fungal pathogens. Its effectiveness depends mainly in the fungal species. *E. floccosum* and *M. canis* are very sensitive, showing significant growth reduction to only 20 ppm ozone during a 10 min exposure (Fig 3). However, other pathogens need longer times at this ozone concentration to show similar leveles of inhibition (*A. terreus* 20 min, *F.solani* and *S. brevicaulis* 30 min). The other three fungal species are even more resistant to ozone, beginning to show significant growth reduction after 30 min of treatment with higher ozone concentration (40 ppm, Fig 3). Ozone, as an oxidant agent, has been used in onychomycosis treatments, and also in other infectious skin conditions [34,36]. Cellular targets for ozone include the membrane lipids, cytosolic enzymes, phenolic and sulfhydryl moieties and double bonds, originating reactive oxygen species (hydrogen peroxide, lipoperoxides, etc.). Fungal cell wall allows transfer of ozone towards inner cell, inhibiting spore germination and biomass formation [35,40].

In this work, the simultaneous treatment with both lasers wavelengths has been combined with the oxidative stress caused in fungal mycelium by the presence of an atmosphere with a high concentration of ozone gas, but using non-toxic concentrations for human skin (40 and 60 ppm), avoiding higher doses (80 ppm) or longer ozone treatment pulses, which may cause skin irritation [36,41,42].

First, 40 ppm ozone treatment was combined with both lasers wavelengths together. This combination was most effective against *T. rubrum*, *M. canis*, *F. solani* and *E. floccosum*, where growth inhibition after 10 min combined treatment was statistically significant in comparison with ozone treatment alone, causing almost total inhibition (Fig 4). *T. mentagrophytes* (20 min

of combined treatment), *A. terreus* (also 20 min) and *A. chrysogenum* (30 min required) are slightly more resistant under this combined treatment conditions. Anyway, in all cases, the combined therapy causes a synergistic effect on viability reduction, compared to only ozone pulse alone, as 40 ppm ozone is able, in most of the cases, to cause total growth inhibition in this pathogens (Fig 4).

In order to compare the effect of a different ozone concentration in combined therapy with lasers pulses, 60 ppm was also used. This ozone concentration, combined with lasers pulses, is able to reduce the time lapse required to achieve the same viability reduction that was obtained with 40 ppm plus lasers treatments (Fig 5). However, *S. brevicaulis* under combined treatment does not show a statistically significant growth reduction in comparison with only ozone (Fig 5).

For the *ex vivo* model experiments, 30 min treatment duration was selected, as this was the most effective time in the combined (lasers plus ozone) *in vitro* experiments. In these conditions (porcine hooves samples), the effectivity is not the same as *in vitro* experiments, and total inhibition was only achieved in the case of *A. chrysogenum* and *E. floccosum*. Growth inhibition was achieved in the cases of *T. ruburm* and *T. mentagrophytes* (Fig 5). The reason of this milder effect may be due to the presence of vegetative mycelium in these experiments, instead of freshly inoculated spores.

In the case of eventual future clinical practice with this method, the patients could be treated several times, on a weekly basis, in order to achieve total eradication of the corresponding onychomycosis. In comparison with current treatments for onychomycosis, which require topical or oral administration of the pharmaceutical drug during months or even years, this method is promising as a fast therapy, lacking also the side effects of most antifungal drugs in the market [21,43].

As a summary, the combined laser (405 nm and 639 nm) and ozone (40 ppm or 60 ppm, which lack toxicity for human skin therapy) treatment developed in this work has demonstrated its usefulness as a fast therapy for onychomycosis *in vitro*, as it is able to inhibit the growth of main dermatophytes and other fungal species involved in this disease. The combination of lasers plus 40 ppm ozone already shows a synergistic effect in comparison with only ozone, and it is very effective against 7 out of the 8 studied fungal pathogens (total inhibition). These results shed light towards a fast and reliable method of clinical practice, without antifungal drugs administration, which may produce side effects in the patient. This combined therapy should be used probably more than once (i.e., on a weekly basis), in order to eradicate totally the corresponding pathogenic fungus in the onychomycosis patient.

## Supporting information

**S1 Rawdata.**
(XLSX)

## Author Contributions

**Conceptualization:** Iván del Valle Fernández, Felipe Lombó.

**Formal analysis:** Javier Fernández.

**Funding acquisition:** Felipe Lombó.

**Investigation:** Javier Fernández, Iván del Valle Fernández, Felipe Lombó.

**Methodology:** Iván del Valle Fernández, Claudio J. Villar, Felipe Lombó.

**Supervision:** Claudio J. Villar.

**Validation:** Javier Fernández.

**Writing – original draft:** Javier Fernández, Claudio J. Villar, Felipe Lombó.

**Writing – review & editing:** Felipe Lombó.

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
