## [Decision Letter · Decision Letter 0]

24 Nov 2020

PONE-D-20-35356

Combined laser and ozone therapy for onychomycosis in an in vitro and ex vivo model

PLOS ONE

Dear Dr. Lombó,

Thank you for submitting your manuscript to PLOS ONE. After careful consideration, we feel that it has merit but does not fully meet PLOS ONE’s publication criteria as it currently stands. Therefore, we invite you to submit a revised version of the manuscript that addresses the points raised during the review process.

There are several points raised by the reviewers that must be addressed. The most important ones are lack of appropriate controls and lack of statistical analysis.

We look forward to receiving your revised manuscript.

Kind regards,

Michael R Hamblin

Academic Editor

PLOS ONE

Journal Requirements:

2. Please state in your methods whether specific permission to use pig hoof samples in research was obtained from the slaughterhouse stated in your methods.

"Authors acknowledge the funding of this work by the Programa Ayudas a Empresas para la Ejecución de Proyectos de I+D+i en el Principado de Asturias en el Periodo 2014-2015 (IE-14-084) and the Programa de Ayudas a Grupos de Investigación del Principado de Asturias (IDI/2018/000120)."

We note that one or more of the authors are employed by a commercial company: Termosalud SL

(2) Please also provide an updated Competing Interests Statement declaring this commercial affiliation along with any other relevant declarations relating to employment, consultancy, patents, products in development, or marketed products, etc.  

4. Please ensure that you refer to Figure 2 in your text as, if accepted, production will need this reference to link the reader to the figure.

Reviewers' comments:

Reviewer's Responses to Questions

**Comments to the Author**

1. Is the manuscript technically sound, and do the data support the conclusions?

Reviewer #1: Yes

Reviewer #2: No

2. Has the statistical analysis been performed appropriately and rigorously? 

Reviewer #1: I Don't Know

Reviewer #2: No

3. Have the authors made all data underlying the findings in their manuscript fully available?

Reviewer #1: Yes

Reviewer #2: Yes

4. Is the manuscript presented in an intelligible fashion and written in standard English?

Reviewer #1: Yes

Reviewer #2: No

5. Review Comments to the Author

Reviewer #1: The manuscript is very well organized, the results are promising; however, minor revision is needed before publication.

1. Please add more up-to-date references in the introduction, and cite them appropriately.

2. I cannot see any statistical analysis in this study, please add the statistical analysis for all of your data

3. please compare your results with the literature in the discussion

4. The figures are really poor, and are not sharp. I cannot follow the trends in the figure. please upload high quality images.

Minor revision

Reviewer #2: The authors propose to use a combination of 405 nm and 630 nm light for the treatment of onychomycosis in vitro and ex vivo.

General comments.

The authors need to address my comments before considering publication.Important controls are missing from the study – namely, treatments using light alone. In addition, it would be ideal to evaluate effects of each light wavelength alone (with and without ozone) to understand the role (and potential complementation) of each light wavelength. Also, a serious concern I have is how the fungal inhibition was quantified – the results appear to be qualitative rather than quantitative – thus no statistical significance between each group of tests can be evaluated.

Specific comments

Line 92 – please can you add a citation showing NADPH-oxidase as the target for 405 nm light in fungi? What about porphyrins?

Line 100 – the use of ozone appears to come out of the blue, I think it would be better to introduce ozone first rather than just say that studies are required to validate efficacy of laser treatment with or without ozone.

Line 102 – what treatments? Also please define ppm when first used in text.

Line 113 – is this the irradiance /cm2? If not, what area? Also, this seems awfully high and I would not classify it as LLLT. Have you measured the thermal effects?

Line 115: have you measured transmittance of light through the bag? If so, please include.

Line 116: please include the emission spectra

Line 131: on or inside the bag? I assume they should be placed inside. Also, were the lids placed on top? If so, how does the lid and plastic influence transmittance?

Line 133: I would think experiments should be performed at least in triplicate (independently over separate days).

Line 137: why were the control samples incubated for only 48 hours and the treated for 5 days? What is the growth rate of each fungal organism? Would you expect these treatments to attenuate growth rather than inhibit/kill.

Line 139: what was the total radiant exposure of each light wavelength that was exposed? If the irradiance is 1.8 (J/cm2?) this would be a very high radiant exposure 1080-3240 J/cm2. I would be concerned about thermal effects.

Line 151:was potassium hydroxide not used to dissolve nail keratin and observe the infected nail?

Line 170: table 1A and B: this method of ‘quantification’ is not easy for the reader to interpret. What does + vs. ++ mean? We know that +++ means indistinguishable from the control but it is not very helpful. I wonder why did the authors not spread the conidia onto the plate to achieve single colonies so that the CFU might be quantified? To me, this seems far too subjective. In addition, How did the authors determine percent inhibition without quantification?

Line 184: I am struggling with the fact that the authors only used the combination of 405 nm and 630 nm under in vitro conditions. For an appropriately designed experiment, they should have also evaluated each respective light wavelength alone to better understand the contribution of each wavelength to the inhibition. The role of the 630 nm wavelength (as per the authors suggestion) is for immunomodulation purposes and to increase blood circulation. In vitro or ex vivo, this is not going to occur. However, it is feasible that 630 nm light might influence the fungal organisms themselves to increase the susceptibility to 405 nm light. Some discussion is needed.

Line 186: why were only 40 ppm and 60 ppm selected. It seemed from the ozone only results, they were very effective alone. it seems only A. terreus was tolerant to this. Ideally, some controls evaluating the role of each wavelength alone should be included. In addition, it seems like light alone (without ozone) are missing?

Line 214: what is know about immune cell trafficking into the keratin?

line 240: do the authors hypothesize that light is an adjuvant for ozone or ozone is an adjuvant for light?

6. PLOS authors have the option to publish the peer review history of their article (what does this mean?). If published, this will include your full peer review and any attached files.

Reviewer #1: No

Reviewer #2: No

---

## [Author Response · Author response to Decision Letter 0]

5 May 2021

Dear Editor,

Please find enclosed the manuscript entitled “COMBINED LASER AND OZONE THERAPY FOR ONYCHOMYCOSIS IN AN IN VITRO AND EX VIVO MODEL” which is re-submitted to “PLOS ONE” (reference PONE-D-20-35356). This work describes a strong effective novel method for treating onychomycosis infections for the main pathogenic fungi in humans. The developed combined therapy uses two laser wavelengths and ozone gas. 

We have accomplished all requirements asked by the referees in the previous revision.

Answers to referees’ comments:

Editor’s comments:

4. Please ensure that you refer to Figure 2 in your text as, if accepted, production will need this reference to link the reader to the figure.

-Figures 2 and 3 have been substituted by three new figures, which now include the statistical analysis. 

Reviewer #1: The manuscript is very well organized, the results are promising; however, minor revision is needed before publication.

1. Please add more up-to-date references in the introduction, and cite them appropriately.

-14 references have been updated.

2. I cannot see any statistical analysis in this study, please add the statistical analysis for all of your data

-Statistical analyses have been carried out and 4 new figures with these analyses have been included in the new version (replacing 2 figures and Tables 1 and 2 in the former version).

3. please compare your results with the literature in the discussion

-Our results have been compared with recent papers in the discussion section.

4. The figures are really poor, and are not sharp. I cannot follow the trends in the figure. please upload high quality images.

-Figures quality depends on the PDF conversion generated in the journal webpage during files uploading. We have enclosed high quality TIFF formats in this version, but they will be transformed as well to PDF format.

Minor revision

Reviewer #2: The authors propose to use a combination of 405 nm and 639 nm light for the treatment of onychomycosis in vitro and ex vivo.

The authors need to address my comments before considering publication. Important controls are missing from the study – namely, treatments using light alone. In addition, it would be ideal to evaluate effects of each light wavelength alone (with and without ozone) to understand the role (and potential complementation) of each light wavelength. Also, a serious concern I have is how the fungal inhibition was quantified – the results appear to be qualitative rather than quantitative – thus no statistical significance between each group of tests can be evaluated.

-We have included in this new version, new experiments with treatments using each laser alone and together. Statistical analysis has been carried out in this new version. 

Specific comments

Line 92 – please can you add a citation showing NADPH-oxidase as the target for 405 nm light in fungi? What about porphyrins?

-We have included a new reference (Huang et al., 2018) and another one for porphyrins effects (den Hollander et al., 2015).

Line 100 – the use of ozone appears to come out of the blue, I think it would be better to introduce ozone first rather than just say that studies are required to validate efficacy of laser treatment with or without ozone.

-We have changed the position of this paragraph, were ozone is described as antimicrobial, also as onychomycosis treatment. 

Line 102 – what treatments? Also please define ppm when first used in text.

-These have been done.

Line 113 – is this the irradiance /cm2? If not, what area? Also, this seems awfully high and I would not classify it as LLLT. Have you measured the thermal effects?

-This is the irradiance per 110 cm2 (the surface area). No thermal effects are detected in the chamber: chamber temperature is always below 30 ºC. 

Line 115: have you measured transmittance of light through the bag? If so, please include.

-The transmittance has been measured and added in the M&M section. 

Line 116: please include the emission spectra

-Data for both emission spectra have been included, showing the minimum, typical and maximum wavelength in each case.

Line 131: on or inside the bag? I assume they should be placed inside. Also, were the lids placed on top? If so, how does the lid and plastic influence transmittance?

-This has been clarified. Inside the bag, without the lids.

Line 133: I would think experiments should be performed at least in triplicate (independently over separate days).

-This has been clarified: triplicates were carried out (not duplicates). 

Line 137: why were the control samples incubated for only 48 hours and the treated for 5 days? What is the growth rate of each fungal organism? Would you expect these treatments to attenuate growth rather than inhibit/kill.

-Controls were incubated the same days than the experiments: 7 days. At this time point (7 days) all species (control plates) show good growth rates on the solid media: from 12 mm until 37.5 mm diameter, depending on the fungal species.

Line 139: what was the total radiant exposure of each light wavelength that was exposed? If the irradiance is 1.8 (J/cm2?) this would be a very high radiant exposure 1080-3240 J/cm2. I would be concerned about thermal effects.

-The actual value is 1.8 W/110.36 cm2, which means 16 mW/cm2. This has been clarified in the manuscript. No thermal effects are detected in the treatment chamber.

Line 151: was potassium hydroxide not used to dissolve nail keratin and observe the infected nail?

-No, in this work NaOH was not used in the ex vivo model. 

Line 170: table 1A and B: this method of ‘quantification’ is not easy for the reader to interpret. What does + vs. ++ mean? We know that +++ means indistinguishable from the control but it is not very helpful. I wonder why did the authors not spread the conidia onto the plate to achieve single colonies so that the CFU might be quantified? To me, this seems far too subjective. In addition, How did the authors determine percent inhibition without quantification?

-New experiments have been carried out during this time, and Tables 1 and 2 have been replaced for the new 4 figures, which include the statistical analyses. 

Line 184: I am struggling with the fact that the authors only used the combination of 405 nm and 639 nm under in vitro conditions. For an appropriately designed experiment, they should have also evaluated each respective light wavelength alone to better understand the contribution of each wavelength to the inhibition. The role of the 639 nm wavelength (as per the authors suggestion) is for immunomodulation purposes and to increase blood circulation. In vitro or ex vivo, this is not going to occur. However, it is feasible that 639 nm light might influence the fungal organisms themselves to increase the susceptibility to 405 nm light. Some discussion is needed.

-New experiments have been carried out now, with each of the two lasers independently or together. The statistical analyses of these experiments is included in the new version of the manuscript. 

Line 186: why were only 40 ppm and 60 ppm selected. It seemed from the ozone only results, they were very effective alone. it seems only A. terreus was tolerant to this. Ideally, some controls evaluating the role of each wavelength alone should be included. In addition, it seems like light alone (without ozone) are missing?

-The reason is that 20 ppm was poorly effective. 40 and 60 ppm were already quite effective. This allowed to avoid the use of the high dose option (80 ppm), potentially preventing future skin irritations due to the pro-oxidant effect of 80 ppm ozone. Another extra reason is that we wished to detect some potential synergistic effect after including lasers in the ozone treatment. The objective was to select the shorter and less potent treatment option. 

Line 214: what is known about immune cell trafficking into the keratin?

-Blood circulation is enhanced in the subungual bed. 

line 240: do the authors hypothesize that light is an adjuvant for ozone or ozone is an adjuvant for light?

-Our new results indicate that light alone is inactive, but it enhances the ozone effect in the combined treatments.

All co-authors have read the report and are in agreement with its content. None of this work has been submitted or published elsewhere.

 Competing Interest

The authors have read the journal’s policy and have the following competing interests: IVF is paid employee of Termosalud SL. There are no patents, products in development or marketed products associated with this research to declare. This does not alter our adherence to PLOS ONE policies on sharing data and materials. All other authors declare that they have no competing interests. 

Financial Disclosure

This study was supported by the Programa Ayudas a Empresas para la Ejecución de Proyectos de I+D+i en el Principado de Asturias en el Periodo 2014-2015 (IE-14-084) and the Programa de Ayudas a Grupos de Investigación del Principado de Asturias (IDI/2018/000120) in the form of grants to FL. IVF received salary from Termosalud SL. The specific roles of these authors are articulated in the ‘author contributions’ section. The funders had no role in the study design, data collection and analysis, decision to publish, or preparation of the manuscript

We thank you for your consideration of this work and we look forward to receiving the new evaluations of this paper. 

Sincerely, 

Felipe Lombó PhD.

Associate Professor of Microbiology

University of Oviedo, SPAIN

email: lombofelipe@uniovi.es

---

## [Decision Letter · Decision Letter 1]

17 Jun 2021

Combined laser and ozone therapy for onychomycosis in an in vitro and ex vivo model

PONE-D-20-35356R1

Dear Dr. Lombó,

We’re pleased to inform you that your manuscript has been judged scientifically suitable for publication and will be formally accepted for publication once it meets all outstanding technical requirements.

Kind regards,

Michael R Hamblin

Academic Editor

PLOS ONE

Additional Editor Comments (optional):

Reviewers' comments:

Reviewer's Responses to Questions

**Comments to the Author**

1. If the authors have adequately addressed your comments raised in a previous round of review and you feel that this manuscript is now acceptable for publication, you may indicate that here to bypass the “Comments to the Author” section, enter your conflict of interest statement in the “Confidential to Editor” section, and submit your "Accept" recommendation.

Reviewer #1: (No Response)

2. Is the manuscript technically sound, and do the data support the conclusions?

Reviewer #1: (No Response)

3. Has the statistical analysis been performed appropriately and rigorously? 

Reviewer #1: (No Response)

4. Have the authors made all data underlying the findings in their manuscript fully available?

Reviewer #1: (No Response)

5. Is the manuscript presented in an intelligible fashion and written in standard English?

Reviewer #1: (No Response)

6. Review Comments to the Author

Reviewer #1: (No Response)

7. PLOS authors have the option to publish the peer review history of their article (what does this mean?). If published, this will include your full peer review and any attached files.

Reviewer #1: No

---

## [Editor Report · Acceptance letter]

21 Jun 2021

PONE-D-20-35356R1 

Combined laser and ozone therapy for onychomycosis in an *in vitro* and *ex vivo* model 

Dear Dr. Lombó:

I'm pleased to inform you that your manuscript has been deemed suitable for publication in PLOS ONE. Congratulations! Your manuscript is now with our production department. 

Kind regards, 

on behalf of

Dr. Michael R Hamblin 

Academic Editor

PLOS ONE